# Neutrophil-mediated oxidative stress and albumin structural damage predict COVID-19-associated mortality

**Mohamed A Badawy[1†], Basma A Yasseen[1†], Riem M El-Messiery[2†], Engy A Abdel-Rahman[1,3†], Aya A Elkhodiry[1†], Azza G Kamel[1†], Hajar El-sayed[1], Asmaa M Shedra[1], Rehab Hamdy[1], Mona Zidan[1], Diaa Al-Raawi[1], Mahmoud Hammad[4], Nahla Elsharkawy[5], Mohamed El Ansary[6], Ahmed Al-Halfawy[7], Alaa Elhadad[4], Ashraf Hatem[8], Sherif Abouelnaga[4], Laura L Dugan[9], Sameh Saad Ali[1]\***

[1]Research Department, Children's Cancer Hospital, Cairo, Egypt; [2]Infectious Disease Unit, Internal Medicine Department, Faculty of Medicine, Cairo University, Cairo, Egypt; [3]Pharmacology Department, Faculty of Medicine, Assuit University, Assuit, Egypt; [4]Pediatric Oncology Department, National Cancer Institute, Cairo University and Children's Cancer Hospital, Cairo, Egypt; [5]Clinical pathology department, National Cancer Institute, Cairo University and Children's Cancer Hospital, Cairo, Egypt; [6]Department of Intensive Care, Faculty of Medicine, Cairo University, Cairo, Egypt; [7]Department of Pulmonary Medicine, Faculty of Medicine, Cairo University, Cairo, Egypt; [8]Department of Chest Diseases, Faculty of Medicine, Cairo University, Cairo, Egypt; [9]Division of Geriatric Medicine, Department of Medicine, Vanderbilt University Medical Center; and VATennessee Valley Geriatric Research, Education and Clinical Center (GRECC), Nashville, United States

**\*For correspondence:**
sameh.ali@57357.org

†These authors contributed equally to this work

**Competing interest:** The authors declare that no competing interests exist.

**Abstract** Human serum albumin (HSA) is the frontline antioxidant protein in blood with established anti-inflammatory and anticoagulation functions. Here, we report that COVID-19-induced oxidative stress inflicts structural damages to HSA and is linked with mortality outcome in critically ill patients. We recruited 39 patients who were followed up for a median of 12.5 days (1–35 days), among them 23 had died. Analyzing blood samples from patients and healthy individuals (n=11), we provide evidence that neutrophils are major sources of oxidative stress in blood and that hydrogen peroxide is highly accumulated in plasmas of non-survivors. We then analyzed electron paramagnetic resonance spectra of spin-labeled fatty acids (SLFAs) bound with HSA in whole blood of control, survivor, and non-survivor subjects (n=10–11). Non-survivors' HSA showed dramatically reduced protein packing order parameter, faster SLFA correlational rotational time, and smaller S/W ratio (strong-binding/weak-binding sites within HSA), all reflecting remarkably fluid protein micro-environments. Following loading/unloading of 16-DSA, we show that the transport function of HSA may be impaired in severe patients. Stratified at the means, Kaplan–Meier survival analysis indicated that lower values of S/W ratio and accumulated $H_2O_2$ in plasma significantly predicted in-hospital mortality (S/W≤0.15, 81.8% (18/22) vs. S/W>0.15, 18.2% (4/22), p=0.023; plasma $[H_2O_2]$>8.6 μM, 65.2% (15/23) vs. 34.8% (8/23), p=0.043). When we combined these two parameters as the ratio $((S/W)/[H_2O_2])$ to derive a risk score, the resultant risk score lower than the mean (<0.019) predicted mortality with high fidelity (95.5% (21/22) vs. 4.5% (1/22), log-rank $\chi^2$=12.1, p=4.9×10$^{-4}$). The derived parameters may provide a surrogate marker to assess new candidates for COVID-19 treatments targeting HSA replacements and/or oxidative stress.

## Editor's evaluation

This submission is novel since it provides information on the structure changes of albumin in COVID-19.

## Introduction

COVID-19 pandemic continues as a global health crisis while the underlying SARS-CoV-2 virus defies all attempted treatment strategies. While writing this report, there have been more than 135 million confirmed cases including around 3 million deaths worldwide according to the World Health Organization Coronavirus Disease Dashboard (https://covid19.who.int/). Although 50% of cases are reported to be in the 25–64 age group, the percentage of deaths increases dramatically with age, and approximately 75% of deaths are in those aged 65 years and above (COVID-19 Hospitalization and Death by Age | CDC). People in the age groups 30–39 years, 40–49 years, and 50–64 years are 4, 10, and 30 times more likely to die from COVID-19 complications compared to the 18–29 years age group. Nevertheless, molecular and cellular factors contributing to mortality outcome in a homogeneous cohort of patients are not yet clear. Lack of diagnostic markers that predict mortality in COVID-19 patients impedes current efforts to siege the pandemic. It is thus critical to identify prognostic tests that can assess the risk of death in critically ill patients to guide clinical protocols and prioritize interventions. Furthermore, mechanistic clues for determining the underlying molecular factors contributing to the hypercoagulability, inflammation, and cytokine storm have been so far illusive. It is therefore imperative to intensify efforts focusing on understanding the molecular pathophysiology of COVID-19 infection and to identify prognostic markers to guide and prioritize clinical decisions.

Human serum albumin (HSA) is the most abundant constituent of soluble proteins in the circulatory system. HSA has been suggested and used as a diagnostic and prognostic marker of numerous diseases and conditions including ischemia, rheumatoid arthritis, cancer, septic shock, among many others. In addition to its numerous physiological and pharmacological functions including the maintenance of blood/tissue osmotic balance (*Singh-Zocchi et al., 1999*), blood pH, metal cation transport and homeostasis (*Bal et al., 2013*; *Stewart et al., 2003*), nutrients and drug shuttling (*Fujiwara and Amisaki, 2013*; *Wishart et al., 2018*), and toxin neutralization (*Ascenzi et al., 2006*; *Vorum and Honoré, 1996*), HSA is suggested to be a major circulating antioxidant (*Cha and Kim, 1996*; *Loban et al., 1997*). HSA can remarkably bind with a diverse array of drugs and toxins thus controlling their bioavailability and pharmacologic effects (*Fasano et al., 2005*). It has been previously shown that more than 70% of the free radical-trapping capacity of serum was due to HSA (reviewed in *Roche et al., 2008*). Importantly, several reports indicated that inflammation enhances vascular permeability of various tissues to HSA apparently to confer antioxidant beneficial effects against reactive species released by activated neutrophils (*Cross et al., 1994*; *Halliwell, 1988*; *Sitar et al., 2013*). Although currently without direct experimental evidence, neutrophilia-mediated oxidative stress was implicated in the COVID-19 pathology and speculated to exacerbate the inflammatory immune response eventually causing multi-organ failure and death (*Laforge et al., 2020*). We hypothesized that COVID-19-mediated oxidative stress may be differentially reflected in HSA's structure and functions and employed electron paramagnetic resonance (EPR) spin labeling spectroscopy to explore HSA's structural changes in correlation with severity and mortality of critically ill COVID-19 patients.

Spin-labeled fatty acids (SLFAs) are established probes to explore structural and functional changes in albumin by EPR spectroscopy (*Ge et al., 1990*; *Haeri et al., 2019*). This approach relies on the well-studied ability of albumin to strongly and exclusively bind with fatty acids in blood. Albumin has at least seven different specific binding sites for long-chain fatty acids located in different domains within the protein (*Bhattacharya et al., 2000*; *Curry et al., 1999*; *Simard et al., 2006*). Effectively, structural and functional changes in HSA may be assessed through the detection of parallel changes in mobility and binding affinity of SLFAs, in addition to the distribution of the spin labels on the albumin molecule (*Haeri et al., 2019*). EPR spectra of spin labels bound to different domains of the protein provide information on the local fatty acids/protein interactions, which may probe changes in the overall structure of the protein under unfolding or damaging conditions (*Figure 1A*; *Bhattacharya et al., 2000*). Here, we compare changes that occur to the mobility, binding affinity, and distribution of the HSA-bound SLFA in whole blood and plasma from COVID-19 patients in critical care unit relative to those observed in normal healthy individuals.

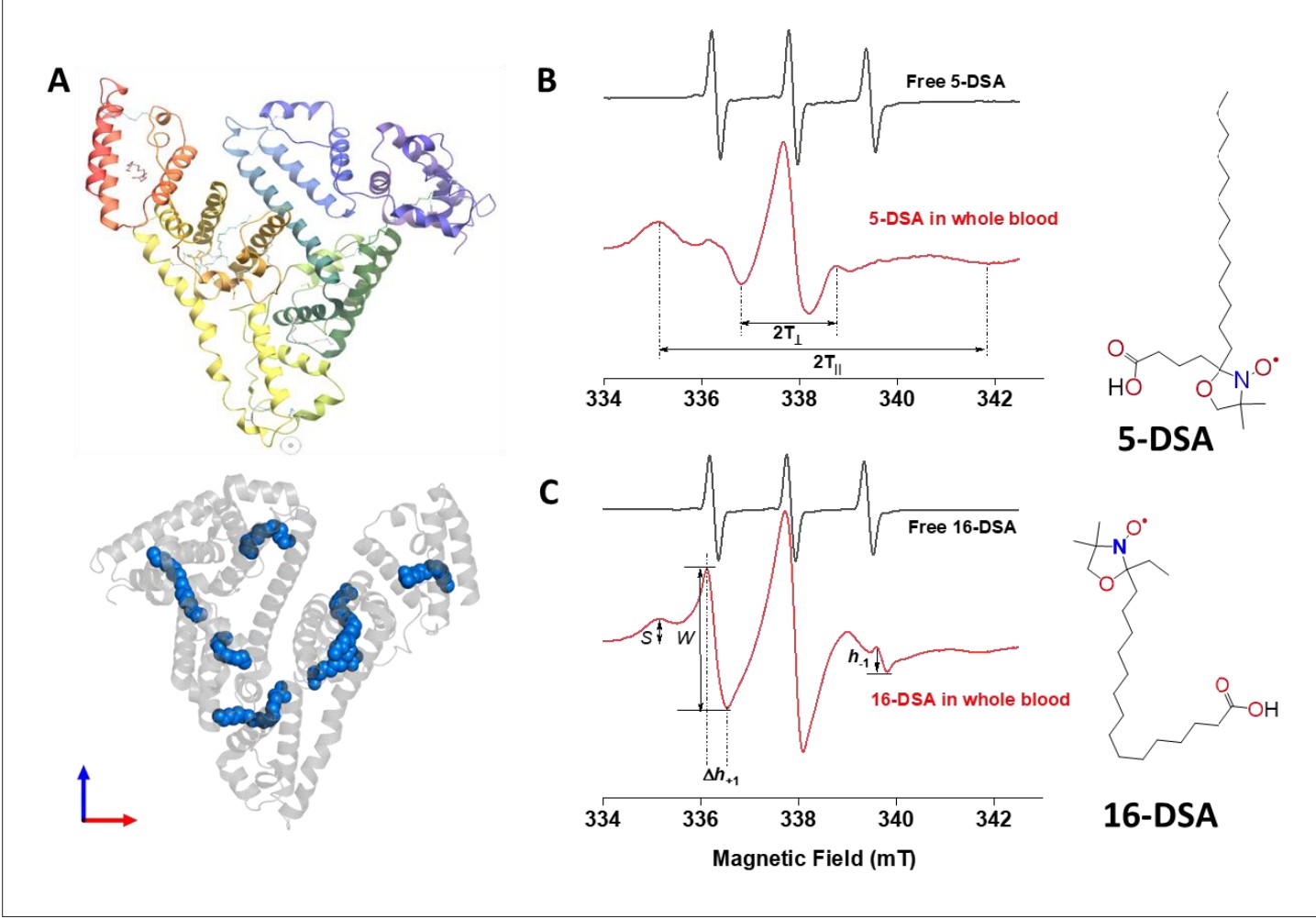

**Figure 1.** Probing structual changes of serum albumin through spin labeling EPR spectroscopy. (**A**) HSA crystal structure containing seven copies of stearic acid. (**B**) Representative EPR spectra of free and HSA-bound 5-DSA (**B**) and 16-DSA (**C**) in whole blood from the same COVID-19 recovered patient. Chemical structures of the two spin-labeled fatty acids are given on the right side of the figure. EPR, electron paramagnetic resonance; HSA, human serum albumin.

## Results

### Demographic, clinical, and laboratory hematologic characteristics of COVID-19 patients

*Table 1* lists demographic data, comorbidities, ongoing medications, and administered anti-COVID-19 medications applied to treat current study participants that were divided into survivors (Sev-R) and deceased (Sev-D). No clinical or demographic characteristic showed statistically significant difference between Sev-R and Sev-D groups when analyzed by Pearson's Chi-square test. In *Table 2*, we show and statistically compare laboratory results of survivors versus non-survivor COVID-19 groups. Although when comparing all parameters in the two COVID-19 groups, we observed changes following the same reported trends in the literature, means' comparisons by Tukey test reported non-significant changes in all parameters except for a significant decrease in albumin level (p<0.05) and a strong trend observed for C-reactive protein (CRP) (greater levels in Sev-D group, p=0.06). Nevertheless, non-survivors' blood carried the frequently observed hallmarks of increased CRP, D-dimer, IL-6, ferritin, and the liver enzymes ALT and AST (reviewed in: *Singh et al., 2021*; *Velavan and Meyer, 2020*). However, it is conceivable that the clinical severe category and the same ICU status of patients in the two groups in addition to relatively small sample sizes underlie the observed lack of robust statistical differences between these parameters.

**Table 1.** Demographic and clinical characteristics of the studied subjects.

|  | Sev-R | Sev-D | Tukey 95% CI | p |
|---|---|---|---|---|
| n | 16 | 23 |  |  |
| Age (mean ± SD) | 60.7±9.5 | 67.8±13.2 | 4.1–17.8 | 0.09 |
| Male | 56.25% | 63.16% |  | 0.677† |
| sO2 (mean ± SD) | 82.1±18.7 | 76.1±18.6 | −20.5 to 8.4 | 0.40 |
| Hypertension | 12.5% | 42.1% |  | 0.053† |
| Diabetes | 25% | 75% |  | 0.08† |
| Cardiovascular disease | 0% | 15.8% |  | 0.10† |
| Cancer | 0% | 10.5% |  | 0.18† |
| Bronchial asthma | 6.25% | 10.5% |  | 0.65† |
| ACE inhibitors | 0% | 7.14% |  | 0.47† |
| ARBs | 9.09% | 7.14% |  | 0.85† |
| calcium channel blocker | 14.28% | 7.14% |  | 0.60† |
| Beta blockers | 0% | 7.14% |  | 0.47† |
| Diuretics | 0% | 7.14% |  | 0.47† |
| Sulphonylurea | 14.28% | 21.43% |  | 0.69† |
| Other oral hypoglycemic | 0% | 21.43% |  | 0.18† |
| Insulin | 42.85% | 21.43% |  | 0.30† |
| Anticoagulant | 57.14% | 50.0% |  | 0.76† |
| Steroids | 71.42% | 64.28% |  | 0.74† |
| Hydroxychloro-quine | 14.28% | 7.14% |  | 0.60† |
| IL-6 receptor antibody | 28.57% | 21.42% |  | 0.72† |
| Proton-pump inhibitor | 28.57% | 42.85% |  | 0.52† |
| Azithromycin | 14.28% | 28.57% |  | 0.47† |
| Cephalosporin | 42.85% | 21.43% |  | 0.30† |
| Carbapenem | 42.85% | 42.85% |  | 1.0† |
| Oxazolidinone | 42.85% | 28.57% |  | 0.51† |
| Fluoro-quinolone | 42.85% | 21.43% |  | 0.30† |
| Nitrofuran | 14.28% | 0% |  | 0.15† |
| Remdesivir | 14.28% | 28.57% |  | 0.47† |
| Ivermectin | 0% | 28.57% |  | 0.11† |

sO2, blood oxygen saturation level; ACE, angiotensin-converting enzyme; ARB, angiotensin II receptor blocker; IL-6, interleukin-6. † p values obtained through Pearson's $\chi^2$ test.

## Neutrophils are a major source of reactive oxygen species

It has been recently proposed that the high neutrophil-to-lymphocyte ratio (NLR) observed in critically ill COVID-19 patients may tip the redox homeostasis due to increased reactive oxygen species (ROS) production (*Laforge et al., 2020*). Our hypothesis implicates elevated oxidative stress as a major cause of HSA damage in severe COVID-19 patients. As a result, we started by following the dependence of clinical outcomes and mortality on ROS levels in blood cells. First, we used flow cytometry to assess percentages of neutrophils, lymphocytes, and platelets in all patients as described in Materials and methods (*Figure 2A*). Furthermore, we used the ROS-sensitive DCF dye to probe intracellular ROS levels in various cell populations in whole blood from all groups. *Figure 2* shows that while

**Table 2.** Laboratory parameters of the current study patients.

WBC, white blood cell; INR, international normalized ratio; CRP, high-sensitivity C-reactive protein; ICU,intensive care unit; PLT, platelet; ALT, alanine transaminase; AST, aspartate transaminase. The Tukey'scalculated p-values as well as upper and lower 95% confidence levels for the Sev-R vs. Sev-D means'comparisons are given.

| | Sev-R (mean ± SD) | Sev-D (mean ± SD) | Tukey 95% CI | p |
|---|---|---|---|---|
| WBCs (×$10^3$ /ml) | 10.6±4.0 | 13.9±8.0 | –1.47 to 8.06 | 0.17 |
| Platelets (×$10^6$ /ml) | 260±75.5 | 213.7±115.8 | –116.7 to 24.2 | 0.19 |
| INR | 1.29±0.56 | 1.23±0.23 | –0.38 to 0.26 | 0.69 |
| CRP (mg/L) | 51.19±54.4 | 103.77±86.3 | –2.35 to 107.5 | 0.06 |
| D-dimer (mg/ml) | 1.47±1.9 | 3.17±3.56 | –0.55 to 3.96 | 0.13 |
| IL-6 (pg/ml) | 314.1±527 | 325.3±619 | –591 to 614 | 0.97 |
| Ferritin | 922.6±565 | 1078±578 | –281 to 594 | 0.47 |
| Albumin (g/ml) | 31.47±7.95 | 26.97±5.1 | –8.7 to –0.26 | 0.038 |
| Hemoglobin (g/dl) | 12.26±2.0 | 12.16±2.0 | –1.53 to –1.33 | 0.97 |
| ALT (U/L) | 33.64±24.15 | 46.5±37.2 | –10.37 to 36.13 | |

ALT, alanine transaminase; AST, aspartate transaminase; CRP, high-sensitivity C-reactive protein; ICU,intensive care unit; PLT, platelet; INR, international normalized ratio; WBC, white blood cell. The Tukey'scalculated p-values as well as upper and lower 95% confidence levels for the Sev-R vs. Sev-D means'comparisons are given.

lymphocyte counts decrease, a parallel dramatic increase in neutrophil counts (% total) was observable when going from Control (40.78±14.0, n=9) to Sev-R (64.0±20.0, n=10) to Sev-D (76.4± 6.8, n=11) groups (overall ANOVA p=$3.9×10^{-5}$). Similar trend was clearly seen in the heat map depicting parameters for all patients analyzed by flow cytometry (*Figure 2B*). It is also clear from *Figure 2B&C* that changes in DCF-positive neutrophils follow similar trend observed for neutrophil counts. To confirm this relation, we compared neutrophil counts with DCF-positive neutrophil counts and found that the two parameters were strongly correlated (Pearson's r=0.8, p=$3×10^{-7}$; *Figure 2C*). Moreover, both parameters individually showed statistically significant increases in both of the studied COVID-19 groups when compared with the control group (*Figure 2D&E*). These results suggest that neutrophils are major sources of elevated oxidative stress in critically ill patients. Note that the observed trends in platelets, lymphocyte, neutrophils, and NLR are similar to reported values (*Sun et al., 2020*; *Yang et al., 2020*).

## Hydrogen peroxide levels in plasma correlate with mortality

Next, we reasoned that elevated oxidative stress in both groups with critical COVID-19 infection would be echoed in plasma levels of hydrogen peroxide. Hydrogen peroxide is the most stable ROS and is highly stable under prolonged storage at low temperatures. We used a highly specific catalase-based assay that we developed and verified in our laboratory to quantify [$H_2O_2$] in plasma samples of all groups. The assay relies on high-resolution detection and quantification of released oxygen due to hydrogen peroxide decompostion by catalase (*Figure 3A&B*). We constructed a calibration curve to confirm the catalase-mediated $H_2O_2$ to $O_2$ stoichiometric conversion (*Figure 3B*). A linear relation was obtained with zero intercept and slope of 0.47±0.03 which closely matches the theoretically expected value of 0.5 (95% confidence interval [CI]: 0.37–0.56, p=$5.6×10^{-4}$, Pearson's r=0.994). Indeed, we detected striking differences between groups even with relatively small sample sizes (Mean ± SD, Control, n=11: 2.95±0.77, Sev-R, n=16: 7.21±2.4, Sev-D, n=23: 9.67±2.0; overall ANOVA p=$2.6×10^{-11}$; *Figure 3C*). The differences between groups have reached statistical significance (Sev-R vs. Control, 95% CI: 2.37–6.13, p=$4.9×10^{-6}$; Sev-D vs. Control, 95% CI: 4.95–8.48, p=0.0; Sev-D vs. Sev-R, 95% CI: 0.90–4.03, p=0.001). It appears from these results that a measure of oxidative stress, that is, [$H_2O_2$] in plasma, is doubled in survivors and tripled in deceased COVID-19 patients relative to controls' plasma average levels.

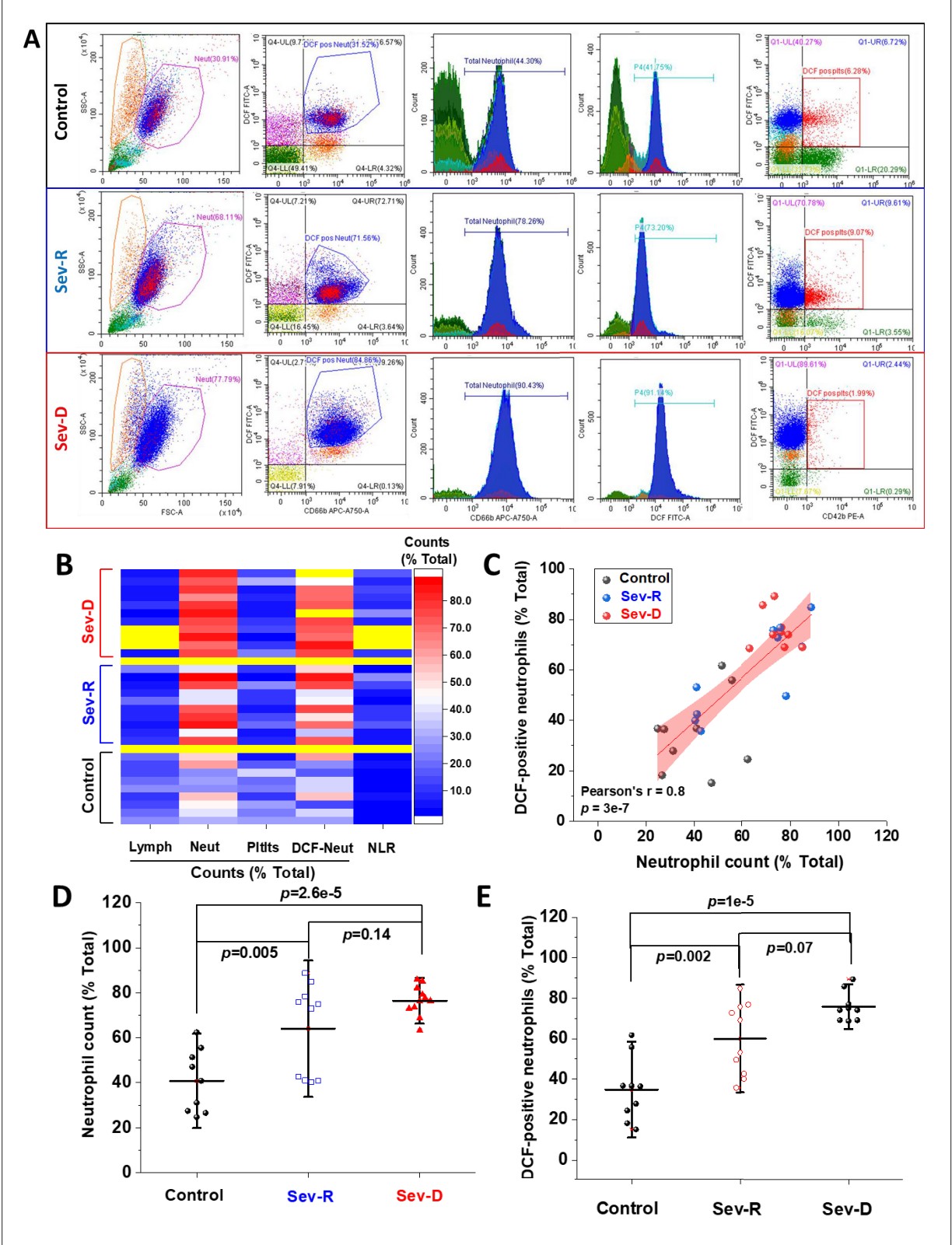

**Figure 2.** Hematologic cellular counts and neutrophil-ROS levels reflect severity and mortality in COVID-19 patients. (**A**) Representative flow cytometric diagrams comparing morphologic, hematologic, and ROS levels in control (representative of n=9; upper row), Sev-R (representative of n=10; middle row), and Sev-D (representative of n=11; lower row) groups. (**B**) Heat diagram comparing lymphocyte, neutrophils, platelets, and DCF-positive neutrophil counts as the percentage of total cell counts in all of the studied subjects. Yellow areas are either group separators or missing data due to

*Figure 2 continued on next page*

*Figure 2 continued*

insufficient sample size or processing errors. (**C**) A diagram showing statistically positive correlation between neutrophil count and count of neutrophils stained positive for DCF dye in all groups (black dots denote controls; blue are Sev-R; and red represent Sev-D patients). (**D**) When neutrophil counts were compared for all groups, both Sev-R and Sev-D groups showed statistically significant neutrophilia relative to control groups. However, only a weak trend has been observed when comparing the two groups with COVID-19. (**E**) DCF staining revealed increased levels of ROS in Sev-R and Sev-D groups relative to control neutrophils. Sev-D showed a trend of increased ROS level relative to Sev-R group. Multiple comparisons were carried out using ANOVA followed by Tukey test and p values are given. ROS, reactive oxygen species.

The online version of this article includes the following source data for figure 2:

**Source data 1.** Raw source data for *Figure 2B-E*.

To confirm this finding, we performed DCF fluorescence imaging on freshly isolated neutrophils of representative group of individuals from each group (*Figure 3D*). We simultaneously stained neutrophils' nuclei with Hoechst 33342 (blue stain) to follow nuclear morphologic changes and DNA diffusion in all groups. Although requiring more detailed studies, a closer look at the acquired images of

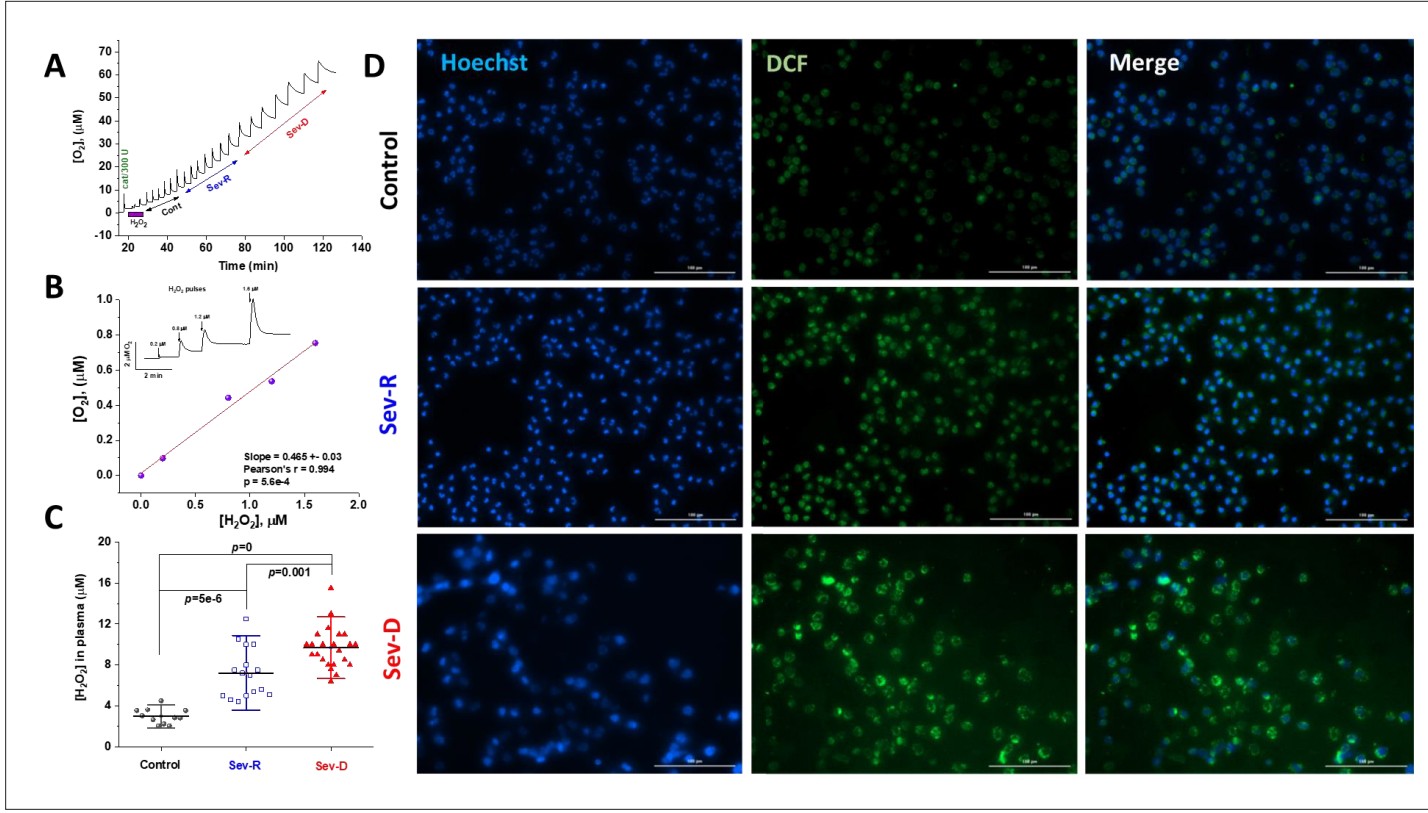

**Figure 3.** Hydrogen peroxide levels in plasma and neutrophils reflect mortality in COVID-19 patients. Catalase was used to specifically and quantitatively determine levels of hydrogen peroxide in identical plasma volumes collected from control (n=11), Sev-R (n=16), and Sev-D (n=23) groups. (**A**) Oxygen levels are monitored and recorded while 50 μl batches of plasma from control, Sev-R, and Sev-D subjects are sequentially infused into tightly air-controlled O2k chamber containing catalase (315 units/ml) in deoxygenated buffer. In addition to the initial rise due to residual oxygen in the added plasma samples, the decomposition of hydrogen peroxide in these samples produces oxygen quantitatively. (**B**) To verify the assay we measured the released oxygen upon adding an increasing volume of standard hydrogen peroxide solution in PBS buffer with 0.2, 0.8, 1.2, and 1.6 μM final concentrations; inset. Linear fitting of the plotted [$O_2$] versus [$H_2O_2$] relation yielded a slope=0.47±0.03 (Pearson's r=0.994, p=5.6×10$^{-4}$), which is very close to the theoretically expected value of 0.5 as the catalase-mediated decomposition of one mole of $H_2O_2$ produces ½-mole $O_2$. (**C**) Plasma contents of $H_2O_2$ in plasma significantly increased in the order Sev-D>Sev-R>Cont using ANOVA followed by Tukey test applied on n=11, 16, and 23 for control, survivors, and non-survivors, respectively. (**D**) Fluorescence imaging was used to assess levels of ROS in freshly isolated neutrophils using DCF (2,7-Dichlorodihydrofluorescein diacetate, green) staining in all groups. Hoechst binds strongly to adenine–thymine-rich regions in DNA thus mapping nuclei through emitting blue fluorescence. Merged DCF and Hoechst images are shown in the third column. Images were acquired using Cytation 5 Cell Imaging Multi-Mode Reader (Agilent) and analyzed using Gen5 Software package 3.08. Scale bar: 100 μm.

The online version of this article includes the following source data for figure 3:

**Source data 1.** Raw polarographic data for released oxygen (A), calibration curve (B), and calculated plasma hydrogen peroxide levels in all groups (C).

Hoechst-stained neutrophils from a survivor patient showed significantly reduced average neutrophil size (control DNA area, 144.8±93.7 $\mu m^2$ [133 cells analyzed] vs. Sev-R DNA area, 36.0±7.5 $\mu m^2$ [241 cells analyzed], Welsh corrected two samples t-test, p=0) with tendencies toward more condensed, more segmented nuclei and frequent C-shaped chromatin. However, neutrophils from a non-survivor exhibited diffused chromatin and possessed in average, a 26% larger DNA area than normal cells and roughly five times that of Sev-R neutrophils (DNA area [97 cells analyzed], 182.7±171 $\mu m^2$, p=0 vs. both control and Sev-R groups). Inspection of the DCF fluorescence images indicated that the non-survivor's neutrophils contained larger populations of what appears to be toxic granules and cytoplasmic vacuoles that are highly ROS-positive. Analysis of mean DCF fluorescence intensities (MFI) per cell confirmed results obtained by flow cytometry and catalase assay reporting increased levels of ROS in the order control<< Sev-R<Sev-D (DCF MFI ± SD [×10$^3$]: Control [282 cells analyzed], 6.7±0.7; Sev-R [351 cells analyzed], 10.9±0.9; Sev-D [368 cells analyzed], 13.0±1.1, Welsh corrected two samples t-test, p=0 for all comparisons).

## EPR-determined biophysical parameters reflecting albumin conformational changes are consistent predictors of COVID-19 mortality

It has been previously shown that non-survivor COVID-19 patients exhibit mild but consistent hypo-albuminemia relative to survivors (*Violi et al., 2020*). We started by assessing albumin levels in the studied cohort of subjects to confirm if they follow similar trends. We found that [albumin] in plasma decreased in the order Control>Sev-R>Sev-D (Mean ± SD, Control, n=8: 40.45±10.93 mg/ml, Sev-R, n=16: 31.47±7.95 mg/ml, Sev-D, n=23: 26.97±5.10 mg/ml; overall ANOVA p=2.4×10$^{-4}$; *Figure 4A*). We detected statistically significant decrease in [albumin] in plasma of Sev-R and Sev-D groups (Sev-R vs. Control, 95% CI: −16.66 to −1.28, p=0.02; Sev-D vs. Control, 95% CI: −20.76 to −6.18, p=1.5×10$^{-4}$; Sev-D vs. Sev-D, 95% CI: −10.28 to 1.28, p=0.15). The reference range of HSA concentration in serum is approximately 35–50 mg/ml, but we and other groups found that COVID-19 associated mortality correlates with lowered [albumin] <30 mg/ml (*Violi et al., 2020*). However, high prevalence of hypoalbuminemia in numerous disease states and the age/sex-dependent wide dynamic range of this protein concentration limits its diagnostic utility (*Levitt and Levitt, 2016*). As a result, we investigated biophysical parameters pertaining to HSA protein configuration in whole blood and plasma of all groups as reflectors of this critical protein functions.

Previous studies reported that allosteric changes in HSA may be utilized to reflect critical functional changes in albumin and explored diagnostic and prognostic values of these changes in cancer (*Haeri et al., 2019*; *Kazmierczak et al., 2006*). It has also been found that long-chain fatty acids binding alters the interactive binding of ligands in the two major drug binding sites of HSA (*Yamasaki et al., 2017*). We employed EPR spectroscopy to probe SLFAs' binding statuses and protein configuration in all groups as detailed in the Materials and methods section above. Analysis of the 5-DSA and 16-DSA EPR spectra revealed remarkable changes in spectral features between control and COVID-19 groups (*Figure 4B* and *Figure 5*, 5-DSA; *Figure 4*, 16-DSA in whole blood). For example, the hyperfine coupling tensor element $2T_{\parallel}$ (defined in *Figure 1B* and used to calculate the protein packing order parameter S) is sensitive to microenvironmental effects (such as polarity, H-bonding, electrostatic interactions, etc.) on the spin probe that is localized in one of the HSA native fatty acids binding pockets. Also, the S/W ratio (see *Figure 1C*) corresponding to strongly bound/weakly bound populations of 16-DSA spin probe may reflect changes in protein folding that can alter protein-fatty acid interactions. Furthermore, the rotational correlation time $\tau_c$ which is a measure of the spin probe rotational mobility is also analyzed.

Calculated EPR spectral parameters for all groups are listed in *Supplementary file 1* including exact ANOVA and Tukey test p values along with the number of subjects analyzed. Both 5-DSA and 16-DSA were used to probe the degree of local interactions in sites where the SLFA is buried in the protein interior (high mobility, 16-DSA) and closer to the protein-aqueous interface (low mobility due to interactions with water molecules and polar amino acids, 5-DSA; *Gantchev and Shopova, 1990*). Indeed, 5-DSA reflected significantly greater S parameter values relative to 16-DSA both in plasma and in whole blood (*Figure 4D*). However, independent of the spin probe and both in plasma and whole blood samples, the order parameter has been consistently lower in Sev-R which was further

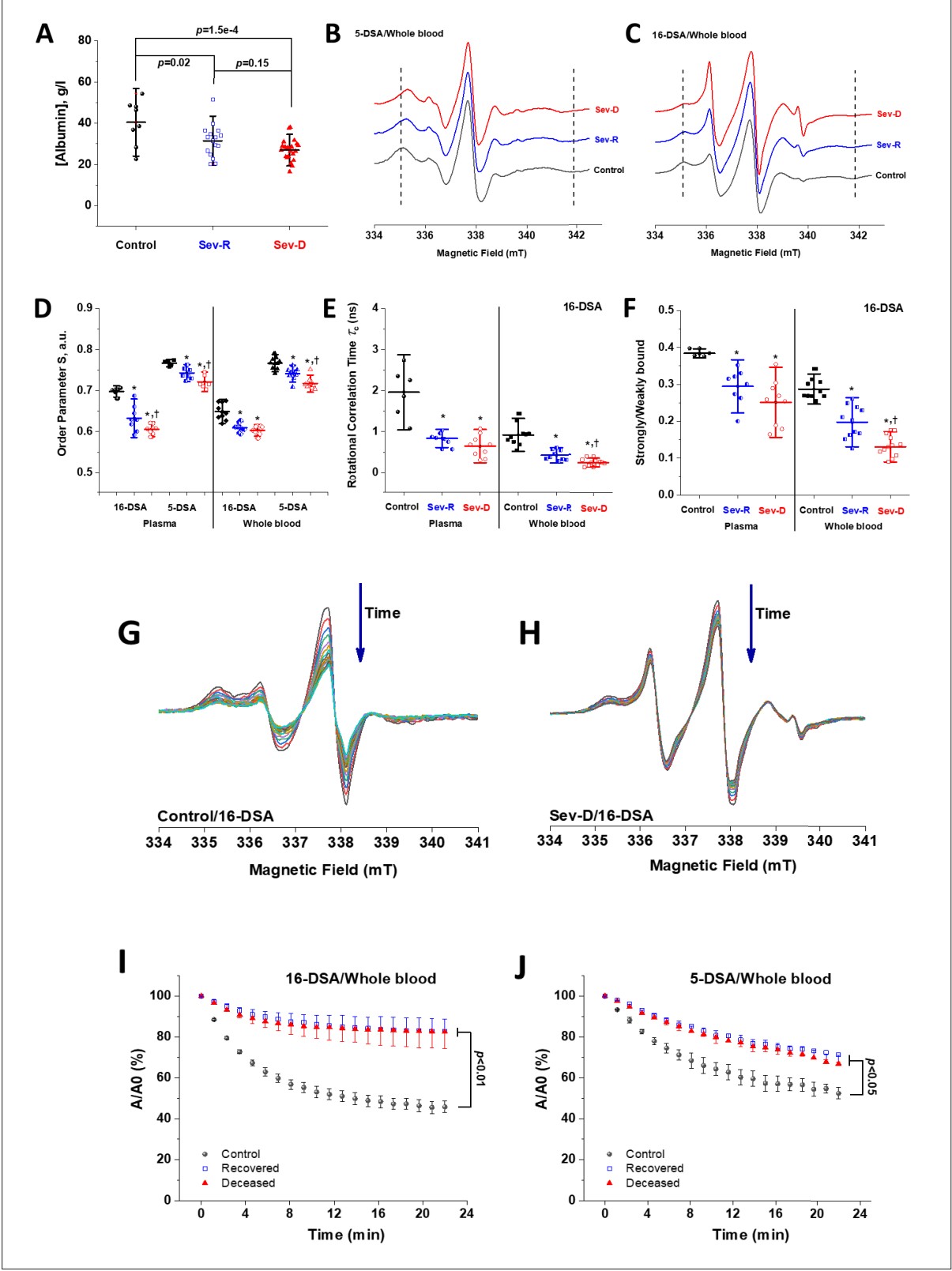

**Figure 4.** EPR spectroscopic analyses of HSA-fatty acid binding reveal strong dependence of binding on mortality in COVID-19 patients. (**A**) Albumin level in plasma of control (n=8), Sev-R (n=16), and Sev-D (n = 23) groups showed that both survivors and non-survivors COVID-19 patients exhibit statistically significant hypoalbuminemia. Comparisons between representative spectra showing changes in line shape that are related to mobility and microenvironmental statuses of HSA-bound 5-DSA (**B**) and 16-DSA (**C**) in whole blood of a control (black trace), a Sev-R (blue trace), and a Sev-D

*Figure 4 continued on next page*

*Figure 4 continued*

(red trace) patients. Calculated biophysical parameters including order parameter (**D**), rotational correlation time (**E**), and the ratio between strongly bound to weakly bound spin labels (S/W, **F**) as defined in *Figure 1* and described in Materials and methods section. Statistical comparisons by ANOVA followed by Tukey tests were used for means' comparisons and revealed remarkable decrease in the binding strengths and packing parameter of the local microenvironment surrounding the spin labels. All calculated parameters along with exact p values are given in the *Supplementary file 1*. Water accessibility into albumin/fatty acids binding pockets are followed by reacting with ascorbate, which reduces nitroxide radicals into the EPR silent hydroxylamines. Representative EPR signal decays of 16-DSA in whole blood of control (**G**) and Sev-D (**H**) are shown. Kinetic traces (n=3 per group) showing the reduction of 16-DSA (**I**) and 5-DSA (**J**) bound to HSA by sodium ascorbate in whole blood. Kinetic traces are shown as the percentage loss of the signal intensity of the middle peak ($A/A_0$). All samples contained 0.26 mM spin label and 3 mM sodium ascorbate and measured at 37°C. Weaker and slower disappearance of the EPR signal suggests inaccessible space toward the nitroxide moiety of the spin label. EPR, electron paramagnetic resonance; HSA, human serum albumin.

The online version of this article includes the following source data for figure 4:

**Source data 1.** Determined biochemical and biophysical EPR parameters used for statistical comparisons.

decreased in Sev-D patients relative to the control group (*Figure 4D*). In whole blood, similar results that showed more statistically robust differences have been observed.

Calculations of $\tau_c$ as described in Materials and methods showed rotational mobility of 16-DSA is significantly faster when bound with HSA from COVID-19 patients relative to that from control subjects in plasma or whole blood. Finally, similar trends have been observed for the S/W parameter

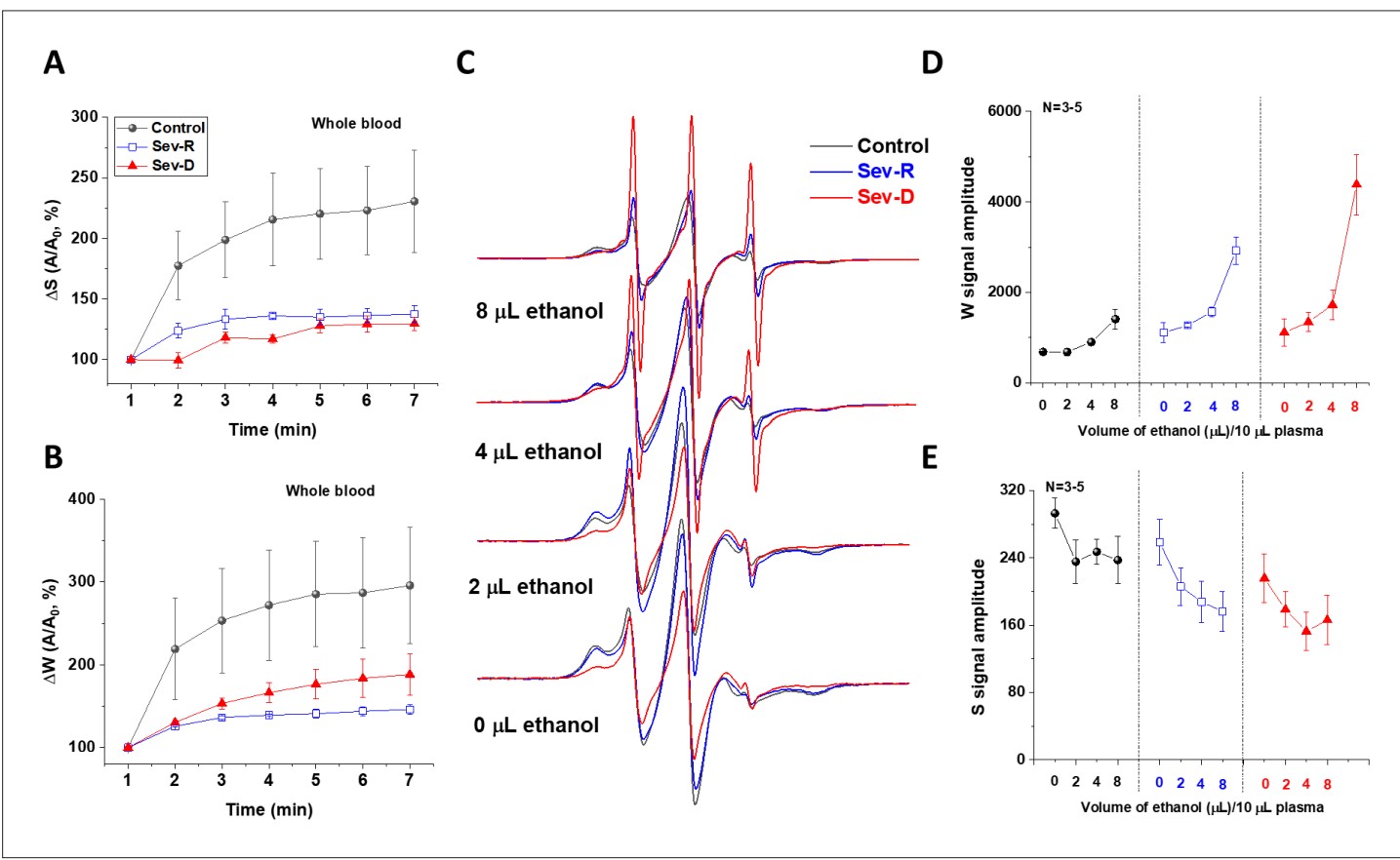

**Figure 5.** COVID-19-associated impairment of HSA transport function. Transport function of HSA is assessed through the apparent kinetics of fatty acid uptake by following the rise in both strongly (**A**) and weakly (**B**) bound components immediately after mixing SLFA with blood from representative subjects (n=3 for each kinetic trace). These results demonstrate the hindered fatty acid uptake by HSA of critically ill patients relative to controls. (**C–E**) To investigate the dislodging function of HSA, increasing volumes of absolute ethanol were added to identical SLFA-plasma mixtures of all groups (n=3–5) and the EPR spectra were acquired (**C**) to follow weakly (**D**) and strongly (**E**) bound populations of SLFA. Redistributions of the fatty acid populations are noticeable through decreased S and increased W (includes signals of free fatty acid) peaks. This redistribution is remarkably pronounced in critically ill patients reflecting weaker association with, and easier release of fatty acids from HSA in those patients. HSA, human serum albumin; SLFA, spin-labeled fatty acid.

which signifies contributions of the strongly and weakly bound components of 16-DSA in different fatty acids pockets. Taken together, these results indicate that COVID-19 pathology is associated with extensive structural changes in the HSA protein that imply the prevalence of malfunctional derivatives of this critical protein.

## Hampered water accessibility into HSA/fatty acid pockets in whole blood of COVID-19 patients

We followed water accessibility toward deep pockets carrying the spin labels through kinetic analysis of the nitroxide radical EPR silencing by the water-soluble ascorbate anion (*Pavićević et al., 2014*; *Figure 4(G–J)*). Under matching experimental conditions, 16-DSA and 5-DSA/HSA signals in whole blood of control subjects decayed remarkably faster when compared with both Sev-R and Sev-D (n=3 per group, p<0.05). Weaker and slower disappearance of the EPR signal of COVID-19 patients by ascorbate suggests less accessible space toward the nitroxide moiety of the spin label within the protein. It is clear from these data that the HSA of COVID-19 patients is generally less water accessible relative to control ones. However, the core of the HSA of both COVID-19 groups was not significantly different in terms of water accessibility.

## Defective transport function of HSA in critically ill COVID-19 patients

Restricted water/ascorbate accessibility in severe patients relative to controls (*Figure 4I&J*) along with the observed significant changes in the microenvironments surrounding SLFAs are generally viewed to implicate functional damage (*Gantchev and Shopova, 1990*; *Pavićević et al., 2014*; *Junk et al., 2010*; *Muravsky et al., 2009*; *Rehfeld et al., 1978*). In this paradigm, changes in the SLFA's lineshape are taken to indirectly display functional changes in the solution shape of albumin due to spatial rear-rangements of paramagnetic centers in EPR-active, albumin-bound fatty acids. However, to confirm that the observed protein damages are indeed functional determinants for HSA, we performed addi-tional experiments exploring albumin transport functions through loading/unloading of SLFA for representative sets of samples (n=3–5) from all of the studied groups. We argued that if the observed HSA structural damages are functionally relevant, they would affect the ability of HSA to load and dislodge fatty acids into and from their binding sites (*Matthes et al., 2002*). To follow the fatty acid loading efficacy, we followed the apparent kinetics of 16-DSA's S (*Figure 5A*; strongly bound) and W (*Figure 5B*; weakly bound) peaks over 7 min immediately after mixing the spin label with identical plasma volumes from all groups. These results (n=3) demonstrate the hindered fatty acid uptake by HSA of critically ill patients relative to controls.

To investigate the dislodging function of HSA, we monitored the effect of exogenously added ethanol on S and W peaks' proportions in all groups. Namely, we compared the amounts of exog-enous ethanol needed to withdraw the weakly and strongly bound components of the HSA-bound SLFAs in whole blood (*Figure 5D&E*). Under these conditions, the W peak envelops contributions from both weakly bound and the released SLFA. Redistributions of the fatty acid populations are noticeable through decreased S and increased W (includes signals of free fatty acid) peaks. This redis-tribution is remarkably pronounced in critically ill patients reflecting weaker association with, and easier release of fatty acids from HSA in those patients. These findings are taken to indicate impaired transport function of HSA in critically ill patients.

## Association of EPR-determined HSA structural changes with neutrophil counts and measures of oxidative stress

In order to verify our proposed links between oxidative stress and albumin damage, we applied linear correlations between plasma levels of hydrogen peroxide and EPR-calculated parameters pertaining to protein packing order parameter (S), fatty acid mobility ($\tau_c$), and S/W ratio in all groups (n=19–26, *Figure 6A–D*). Furthermore, to substantiate our suggestion that neutrophils are major sources of oxidative stress implicated in COVID-19 severity and mortality, we explored linear correlations between the populations of DCF positive neutrophils and $\tau_c$ (*Figure 6E*) or S/W ratio (*Figure 6F*). Despite relatively small numbers of subjects analyzed, correlations of biophysical parameters showed moderate-to-strong negative correlations with [$H_2O_2$] in plasma or with DCF positive populations (% of total); Pearson's r=–0.6 to –0.78, p<0.01 for all correlations. These observations may suggest that

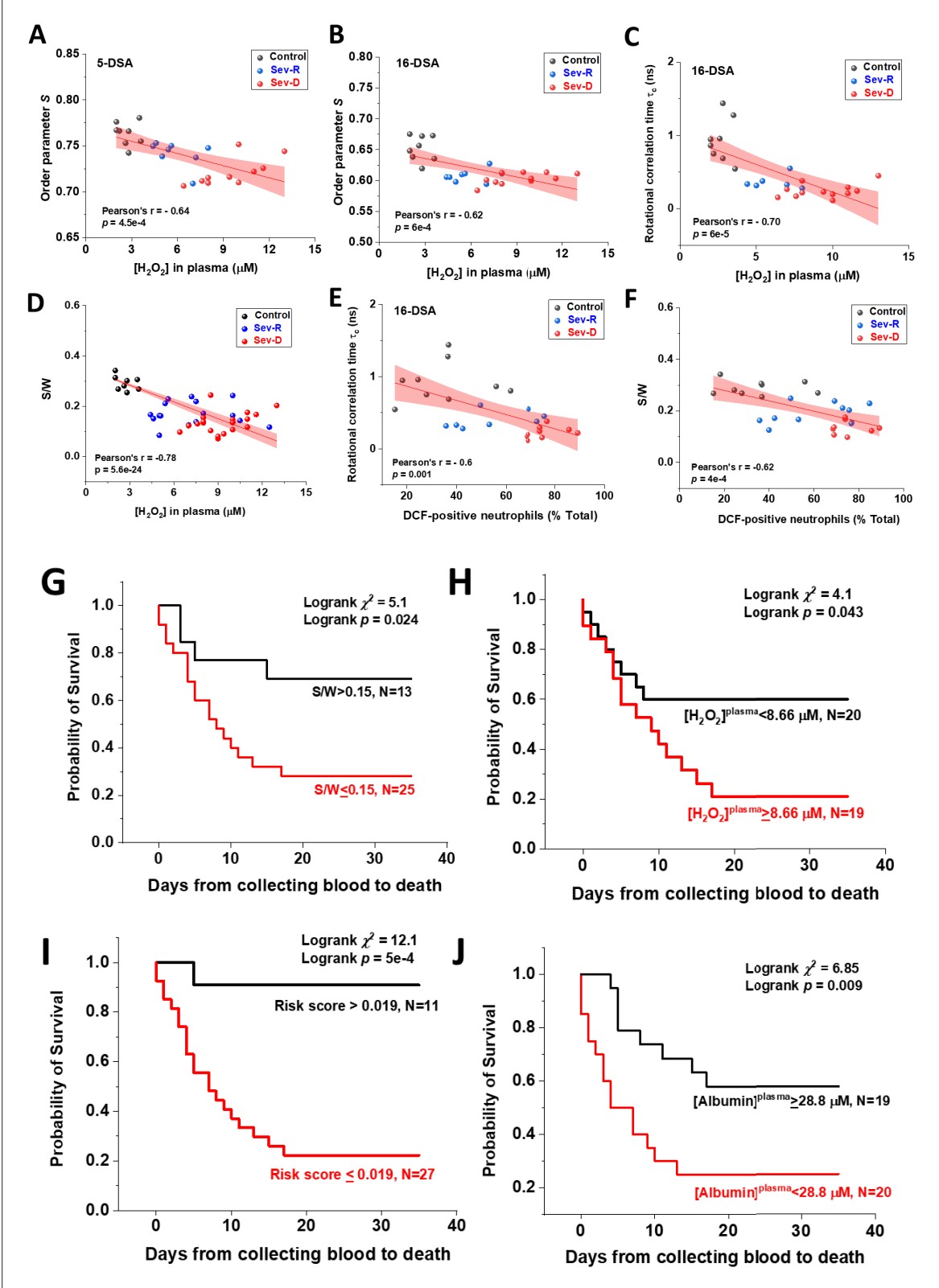

**Figure 6.** Associations of oxidative stress measures and biophysical parameters with mortality outcome in COVID-19 subjects. Linear correlations between plasma levels of hydrogen peroxide and EPR-calculated parameters pertaining to protein packing order parameter (**A, B**), fatty acid mobility $\tau_c$ (**C**), and S/W ratio (**D**) in Control (black circles), Sev-R (blue circles), and Sev-D (red circles) groups (n=26, 27, 26, 46, 26, and 28 for data in panels (**A–F**), respectively). Significant negative correlation between S/W ratio and plasma levels of hydrogen peroxide is seen (**D**). Neutrophils are major sources

*Figure 6 continued on next page*

*Figure 6 continued*

of oxidative stress as evident from the linear correlations between % DCF positive neutrophils and $\tau_c$ (**E**) or S/W ratio (**F**). On each correlation, Pearson's r and exact p values are provided. Kaplan–Meier estimates of time-to-mortality from blood sample collection during ICU hospitalization (**G–J**). Log-rank Kaplan–Meier survival analyses were carried out to estimate the probability of survival of COVID-19 patients in relation to cutoff thresholds arbitrarily selected as the mean values of the analyzed parameters. For S/W, plasma [$H_2O_2$], Risk Score defined as the {(S/W)/[$H_2O_2$]} ratio, and plasma [Albumin], the number of analyzed COVID-19 patients were 38, 39, 38, and 39, respectively. EPR, electron paramagnetic resonance.

The online version of this article includes the following source data for figure 6:

**Source data 1.** Raw data for correlation analyses and for Kaplan-Meier analyses.

## Analysis of Kaplan–Meier estimates of time-to-mortality from blood sample collections during ICU hospitalization

Finally, we stratified patients into two groups, patients showing values below the means and patients with values above the means of each of the studied parameters: (i.e., S/W mean=0.15 [0.071–0.24]; plasma [albumin] mean=28.8 mg/ml [16.41–51.48]; and plasma [$H_2O_2$] mean=8.66 µM [4.4–15.5]). Patients with lower values of the S/W ratio showed significantly higher in-hospital mortality (81.8% vs. 18.2%, log-rank $\chi^2$=5.1, p=0.024, **Figure 6G**). Similarly, patients with accumulated $H_2O_2$ in their plasma showed higher mortality (65.2% vs. 34.8%, log-rank $\chi^2$=4.1, p=0.043, **Figure 6H**). However, when we combined these two parameters to derive a risk score as the ratio ((S/W)/[$H_2O_2$]), the resultant risk score lower than the mean (<0.019) predicted mortality with excellent accuracy (95.5% vs. 4.5%, log-rank $\chi^2$=12.1, p=5×10⁻⁴, **Figure 6I**). This consistent statistics must be verified over larger sample size which is currently in pursuit. Under our current conditions, order parameter (not shown) was not statistically significant determinant of in-hospital mortality. Finally, albumin level in plasma was found to predict mortality with acceptable accuracy (65.2% vs. 34.8%, log-rank $\chi^2$=6.85, p=0.009, **Figure 6J**).

## Discussion

HSA is a pivotal protein with diverse multifaceted functions that are increasingly reported to reflect physiologically and contribute to pathological states (**Levitt and Levitt, 2016**; **Coverdale et al., 2018**). Closer to the context of COVID-19 pathophysiology, critical roles of HSA have been suggested as it acts as an anti-inflammatory and antioxidant protein (**Inoue et al., 2018**), anticoagulant agent (**Doweiko and Nompleggi, 1991**; **Ronit et al., 2020**), inhibitor of oxidative stress-mediated clotting and platelet activation (**Basili et al., 2019**; **Tian et al., 2020**), drug-dependent allosteric carrier and regulator (**Fanali et al., 2012**), and a potent heme scavenger that may also exhibit globin-like reactivity (**Ascenzi et al., 2015**). In the light of these established functions, it is natural to suggest albumin as a frontline protection against COVID-19-associated lethality resulting from cytokine storm, oxidative stress, blood clotting, and the ensuing organ failure. Although hypoalbuminemia has been repeatedly reported as a predictor of mortality in COVID-19 (**Violi et al., 2020**; **Huang et al., 2020**), low albumin levels may result from surgery, dialysis, abdominal infections, liver failure, pancreatitis, respiratory distress, bypass surgery, ovarian problems caused by fertility drugs, and many other conditions. High prevalence of hypoalbuminemia in numerous disease states and the age/sex-dependent wide dynamic range of this protein concentration limits its diagnostic utility (**Levitt and Levitt, 2016**). As a result, we investigated biophysical parameters pertaining to HSA protein structure in whole blood and plasma of all groups as reflectors of this critical protein functions.

We designed the present study to explore how COVID-19-caused mortality is related to oxidative stress and whether ROS-induced albumin damage can be used to predict such outcome. To avoid disparities related to clinical statuses and interventional protocols we restricted our subject recruitment to a homogeneous cohort of patients in terms of diseases severity, hospitalization, and interventions. Both sexes were represented in the overal COVID-19 subjects (40% females) and sex was not found to affect mortality. The mean age of the studied subjects was 66.7±8.9 years (42–81), and non-survivors were slightly orlder. No other clinical or demographic characteristic showed statistically

significant difference between survivors and non-survivors which further indicates the homogeniety of the studied pool of subjects.

Our current data provide converging and novel evidence that neutrophils are major players in oxidative stress associating inflammation in critically ill COVID-19 patients, especially those that do not survive the infection. We detected increased neutrophil count, increased ROS-positive neutrophil population, increased intracellular ROS level in these neutrophils, and a remarkably elevated hydrogen peroxide residue in plasma of non-survivor COVID-19 patients relative to control and survivor groups. Moreover, these oxidative stress measures were found to strongly correlate with EPR-detected structural damages of HSA. We found that albumin levels were inversely associated with overall mortality, and that a biophysical measure of structural changes inflicted on albumin (S/W) can consistently predict overall mortality even with a relatively small n=38 sample size. Moreover, plasma levels of hydrogen peroxide were also found to predict in-ICU mortality (n=39). The current survival analysis reports event numbers in the range from 15 to 21 for various predictor parameters (see new *Figure 6G–J*). This exceeds the current consensus that 10 or more events per predictor are sufficient to provide firm grounds for generalization of findings (*Courvoisier et al., 2011*; *Kocak and Onar-Thomas, 2012*; *Ogundimu et al., 2016*). It is important to mention that the current work did not address important questions related to the enzymatic source of ROS and whether mitochondria contribute to the observed oxidative stress profiles in connection with infection severity and mortality.

Oxidative stress is commonly implicated in a plethora of diseases and was indeed suggested repeatedly; albeit with little experimental support, as the spearhead of inflammation leading to severe symptoms and sepsis-like fatal responses in critically ill COVID-19 patients (*Laforge et al., 2020*). We suggest that HSA damage inflicted by neutrophil-mediated ROS bursts in those patients effectively contributes to the COVID-19 pathology. For instance, impairment of HSA ability to transport an extremely wide range of endogenous and exogenous ligands may affect the bioavailability of adminsitered drugs (*Lee and Wu, 2015*), distribution of anti-inflammatory molecules such as NO and prostaglandins (*Arroyo et al., 2014*), as well as affecting its anticoagulation and antiplatelet activation functions (*Paar et al., 2017*). Reduced and damaged HSA may also disturb the vascular oncotic pressure contributing to pulmonary edema and potentially to gastrointestinal symptoms that characterize recent waves of the infection. Collectively, these factors constitute major players in the pathology triggered by SARS-CoV-2 viral infection. Although hypoalbuminemia is suggested as a pivotal marker for the pathology of many infections (reviewed very recently in *Wiedermann, 2021*) including in COVID-19 (*Violi et al., 2020*), albumin damage has not been directly implicated in infectious disease to the best of our knowledge.

We believe that the current data suggest that albumin-related changes are more robust predictors of mortality and severity than immune-related parameters, such as cytokine levels, CRP, D-dimer, and so on. These parameters were not statistically different when comparing survivors with non-survivors (*Table 2*) . Nevertheless, obtained biophysical and biochemical parameters correlated reasonably well with severity and mortality. Our current findings substantiate the utilization of the identified parameters as accurate mortality predictors of critically ill COVID-19 patients. Determining threshold values of these parameters can possibly help to identify patients in need of urgent medical attention, and may provide novel markers to assess new candidates for COVID-19 treatments targeting HSA replacements. We thus suggest the importance of studying the potential effects of albumin replacement therapy on clinical outcomes of Covid-19 patients. Our results also provide new mechanistic insights into pathways that could be targeted to help prevent critical COVID illness and death, for example, oxidative stress.

# Materials and methods

## Key resources table

| Reagent type (species) or resource | Designation | Source or reference | Identifiers | Additional information |
|---|---|---|---|---|
| Antibody | CD-42b-PE- (Mouse monoclonal) | Beckman Coulter Life Sciences | Cat# IM1417U RRID: AB_2893282 | FACS (1 μl per test) |

*Continued on next page*

| Reagent type (species) or resource | Designation | Source or reference | Identifiers | Additional information |
|---|---|---|---|---|
| Antibody | CD14-PC7- (Mouse monoclonal) | Beckman Coulter Life Sciences | Cat# A22331 RRID: AB_10639528 | FACS (1 µl per test) |
| Antibody | CD66b-APC-Alexa Fluor 750- (Mouse monoclonal) | Beckman Coulter Life Sciences | Cat# B08756 RRID:AB_2893284 | FACS (1 µl per test) |
| Antibody | CD3-ECD- (Mouse monoclonal) | Beckman Coulter Life Sciences | Cat# IM2705U RRID: MGI:3850637 | FACS (1 µl per test) |
| Other | 2',7'-dichlorofluorescein-diacetate | Sigma-Aldrich | Cat# D6883 | Imaging (30 µM) FACS (20 µM) |
| Other | DAPI | Thermo Fisher Scientific | Cat# 62249 | Imaging (6.15 µg/ml) |
| Other | 2-(3-carboxypropyl)-4,4-dimethyl-2-tridecyl-3-oxazolidinyloxy (5–130 doxyl-stearic acid, 5-DSA) | Sigma-Aldrich | Cat# 253618 | EPR (0.26 mm) |
| Other | 2-(14-carboxytetradecyl)-2-ethyl-4,4-dimethyl-3-oxazolidinyloxy (16–131 doxyl-stearic acid, 16-DSA) | Sigma-Aldrich | Cat# 253596 | EPR (0.26 mm) EPR/ethanol experiments (0.52 mm) |

## Study design and participants

The present study aims to analyze HSA protein configuration statuses in the most severe cases of COVID-19 in comparison with control subjects. This is a prospective observational cohort study of patients with confirmed RT-PCR positive COVID-19. Nasopharyngeal swab RT-PCR results and Lung CT scans were combined to classify severe symptomatic COVID-19 cases. All patients were recruited from Kasr Alainy Cairo University Hospital/ICU-facility at the Internal Medicine Quarantine Hospital. Supportive therapy, including supplemental oxygen and symptomatic treatment, was administered as required. Patients with moderate to severe hypoxia (defined as requiring fraction of inspired oxygen [FiO$_2$]≥40%) were transferred to intensive care for further management including invasive mechanical ventilation when necessary. Patients recruited in the current study were divided into two arms based on future mortality outcome: those who survived the past 15 days following blood samples collection (Sev-R) and those who died within 15 days of samples collection (Sev-D).

No power analysis was done due to lack of reported effect size under matching or similar conditions. Sample size was based on sample availability and collection continued until a total of N=40 has been reached (39 samples analyzed in the current study) for COVID-19 patients diagnosed as severely infected with SARS-CoV-2 and admitted to the ICU during the period from October 13, 2020 to February 21, 2021. Within the follow-up time, 23 patients had died. However, in a few cases and due to leukopenia or blood samples insufficiency to run every experiment on each subject, a minimum subject number of 6 were analyzed to infer various biophysical and biochemical parameters. Nevertheless, all available 39 samples were investigated by blinded operators and were all included in the final analysis. Analysis of fresh samples precluded operators' bias as the censored outcome is a future event. Demographic and clinical data of the studied 39 COVID-19 confirmed positive cases were categorized according to mortality and used for correlative analyses *Table 1*. No randomization was done and clinical data were made available after EPR data collection and analysis had been performed. Patients who survived 15 days post sample collections were considered survivors (Sev-R) while those who died within this period were placed in the Sev-D group.

Written informed consents were obtained from all participants in accordance with the principles of the Declaration of Helsinki. For COVID-19 and control blood/plasma collection, Children's Cancer

Hospital's Institutional Review Board (IRB) has evaluated the study design and protocol, IRB number 31-2020 issued on July 6, 2020.

## Electron paramagnetic resonance measurements

We measured EPR spectra at 37°C using a Benchtop Magnettech MiniScope MS5000 spectrometer (now Bruker Biospin, Berlin) equipped with biotemperature control and computerized data acquisition and analysis capabilities. Typical instrumental parameters during these measurements were: microwave frequency 9.47 GHz, microwave power 10 mW, modulation frequency 100 kHz, modulation amplitude 0.2 mT, and magnetic field range 332–342 mT. Each spectrum was the average of 5 scans with scan time of 60 s. Spin-labeled stearic acids (*Figure 1*): 2-(3-carboxypropyl)-4,4-dimethyl-2-tridecyl-3-oxazolidinyloxy (5-doxyl-stearic acid, 5-DSA) and 2-(14-carboxytetradecyl)-2-ethyl-4,4-di methyl-3-oxazolidinyloxy (16-doxyl-stearic acid, 16-DSA) were purchased from Aldrich Chemical Co. (Milwaukee, USA). SLFA-serum albumin complexes were formed as recommended (*Haeri et al., 2019*) in 0.01 M phosphate buffer (pH 7.4). Conditions were optimized to avoid aggregation of free SLFA and to minimize contributions from albumin-unbound SLFA. Signals were stable over the measurement time and at least up to 1 hr afterward as observed by following time evolution of the recorded EPR spectra.

### Analysis of biophysical EPR parameters

Fluidity measurement was carried out as we reported (*Head et al., 2010*) and explained (*A Abdel-Rahman et al., 2016*) previously. Nitroxyl radicals SLFA probes were used to determine local fluidity near the protein/aqueous interface (5-DSA) or the hydrophobic protein cores (16-DSA) of HSA. Whole blood or collected plasma from all subjects were labeled with 5-DSA or 16-DSA. Ethanolic solutions (0.02 M) of the spin labels were added to the blood/plasma in the ratio 1:100 and the cells were measured at 37°C. From the spectrum of 5-DSA an order parameter (S) was derived by measuring the outer and inner hyperfine splitting $2T_\parallel$ and $2T_\perp$ as defined in *Figure 1* (*Gordon et al., 1989*; *Marczak et al., 2006*) using the formula:

$$S = \frac{T_\parallel - T_\perp}{T_{zz} - T_{xx}} \cdot \frac{a_N}{a_N'}$$

where

$$a_N = \frac{1}{3(T_{zz} + 2T_{xx})}; a_N' = \frac{1}{3(T_\parallel + 2T_\perp)}$$

$T_{xx}$ and $T_{zz}$ are principal values of hyperfine tensor T taken as 0.61 mT and 3.24 mT, respectively (*Marczak et al., 2006*; *Sauerheber et al., 1980*).

To estimate fluidity of the inner lipophilic protein compartment, we used 16-DSA, that is, a spin probe containing the nitroxide group attached on C16 that is located on the opposite terminal relative to the charged carboxyl fatty acid terminus. Rotational correlation time ($\tau_c$), which is taken as a measure of microenvironment fluidity in the fatty acid carrier sites in the HSA protein core, was calculated as (*Schreier et al., 1978*):

$$\tau_c = 6.6 \times 10^{-10} \times h_{+1} \left[ \left( \frac{h_{+1}}{h_{-1}} \right)^{\frac{1}{2}} - 1 \right]$$

where the parameters $\Delta h_{+1}$, $h_{+1}$, and $h_{-1}$ are determined from EPR spectra as shown in *Figure 1C*.

16-DSA spectra (*Figure 1C*) were also analyzed to obtain the ratio between weakly (W) and strongly (S) bound subpopulations of the spin label in sterically hollow versus relatively jammed protein microenvironments (*A Abdel-Rahman et al., 2016*; *Marczak et al., 2006*).

## Phenotyping of peripheral blood by flow cytometry

Citrate-anticoagulated whole peripheral blood was incubated with RBCs lysis buffer containing $NH_4Cl$ (ammonium chloride), $NaHCO_3$ (sodium bicarbonate), and EDTA (disodium) for 15 min (*Don-Doncow et al., 2019*). This lysis buffer is known to preserve viability of cells and is the buffer of choice for assessment of neutrophil respiratory burst (*Vuorte et al., 2001*). Moreover, we have suspended the obtained cells after lysis in phosphate-buffered saline (PBS). Lysed blood was then centrifuged at

500×$g$ for 5 min. Cells were washed two times and then resuspended in PBS to avoid using buffers containing FBS or BSA known to cause activation of neutrophils. Distribution status of platelets, neutrophils, monocytes, and lymphocytes was measured in whole blood samples by 13-color flow cytometry as described using CytoFLEX system (Beckman Coulter Life Sciences CytoFLEX benchtop flow cytometer). Suspended cells were incubated with combinations of anti-human monoclonal antibodies for subset identification as follows: CD-42b-PE (Beckman Coulter Life Sciences, IM1417U) for platelets, CD14-PC7 (Beckman Coulter Life Sciences, A22331) for monocytes, CD66b-APC-Alexa Fluor 750 for neutrophils (Beckman Coulter Life Sciences, B08756), and CD3-ECD (Beckman Coulter Life Sciences, IM2705U) for lymphocytes cells were incubated for 30 min in the dark at room temperature. At the end of the incubation period, cells were washed with PBS and resuspended in 300 µl PBS. Samples were then analyzed by flow cytometry for gating platelet-specific CD42b-PE positive population, neutrophil-specific CD66b-APC-Alexa Fluor 750 positive population, monocytes-specific CD14-PE positive population, and lymphocytes specific CD3-ECD positive population. A number of 20,000 events were acquired and analyzed using CytExpert software to determine the percentage and mean fluorescence intensities (MFIs) of analyzed cell subsets.

## Measurement of ROS by flow cytometry

The intracellular ROS generation by different cell populations was measured in whole blood samples using 2′,7′-dichlorofluorescein-diacetate (DCF, Sigma-Aldrich, D6883). Suspended cells were incubated with DCF (20 µM) and combinations of monoclonal antibodies; CD-42b-PE (Platelets), CD14-PC7 (monocytes), CD66b-APC-Alexa Fluor 750 (neutrophils), and CD3-ECD (lymphocytes) for 30 min at room temperature in the dark. Cells were then washed with PBS and resuspended in 300 µl of PBS. A total of 20,000 events were recorded and analyzed using CytExpert program.

## Determination of hydrogen peroxide concentrations in plasma

Catalase was used to specifically and quantitatively determine the levels of hydrogen peroxide in identical plasma volumes collected from the study subjects. Oxygen levels are monitored and recorded while 50 µl batches of plasma from control, Sev-R, and Sev-D subjects are sequentially infused into tightly air-controlled O2k chamber containing catalase (315 units/ml) in deoxygenated buffer. A continuous stream of pure nitrogen gas was blown over the chamber's sealing cap to prevent oxygen diffusion into the working solution. In addition to the initial rise due to residual oxygen in the added plasma samples, the decomposition of hydrogen peroxide in these samples produced oxygen in a quantitative manner, that is, decomposition of one mole of $H_2O_2$ produced ½-mole $O_2$. To verify the assay, we measured the released oxygen upon adding increasing concentrations of standard hydrogen peroxide solution in PBS buffer. Linear fitting of the plotted $[O_2]$ versus $[H_2O_2]$ relation yielded a slope=0.47±0.03 (Pearson's r=0.994, p=5.6×10$^{-4}$), which is very close to the theoretically expected value of 0.5.

## Neutrophils' isolation

Whole blood samples (10 ml) were collected on ACD tubes and transferred to 15 ml polypropylene conical tubes then centrifuged (300×$g$) for 15 min with reduced acceleration and no brakes. The upper layer (PRP) was transferred to a separate tube. The lower layer was layered over equal volume of Histopaque-1077 and tubes were centrifuged at 500×$g$ for 35 min with reduced acceleration and no brakes. After centrifugation, the first three layers were discarded and the layer containing neutrophils was collected and diluted to 10 ml with HBSS, then centrifuged again at 350×$g$ for 10 min. Supernatant was removed and the pellet was suspended in RBCs lysis buffer and incubated for 15 min at room temperature, then centrifuged at 350×$g$ for 10 min as previously described (*Abdel-Rahman et al., 2021*). The supernatant was removed and the pellet was suspended in 100 µl PBS. The count was obtained manually using a hemocytometer. Cellular viability was checked for each sample by trypan blue and cells were not analyzed unless it exhibited more than 90% viability. For the fluorescence imaging using cytation 5, neutrophils were seeded in a flat clear bottom 96-well plate in a density of 100,000 neutrophils/well.

## Measurement of ROS in isolated neutrophils by fluorescence imaging

The intracellular ROS generation of isolated neutrophils was detected and quantified using the ROS-sensitive DCF dye. Isolated cells were seeded and incubated for 30 min at 37°C followed by centrifugation to form a monolayer. Cells were then stained with DCF at a final concentration of 0.3 mM in PBS for 30 min. Cells were washed and stained with Hoechst 33342 Solution (Thermo Fisher Scientific, 62249) at a final concentration of 10 µM in PBS for 30 min at 37°C. Cytation 5 Cell Imaging Multi-Mode Reader (Agilent) was used to acquire images using 20× lens and the proper fluorescence filter cubes ($\lambda_{ex}$=500±12 nm and $\lambda_{em}$=542±14 nm). Images were processed to quantify fluorescence intensity using Gen5 software package 3.08.

## Determination of plasma albumin concentrations

Albumin concentration in plasma was measured using bromocresol green (BCG) dye as the color intensity of the complex formed is proportional to albumin concentration. Standard curve was generated using standard albumin and samples were measured in 96-well plates at 630 nm using Cytation 5 (Agilent). Concentrations were calculated using the equation generated from standard curve and presented in mg/ml.

## Statistical analysis

Statistical analysis and data graphing were performed using OriginPro 2017 (OriginLab Corporation, Northampton). Data outliers were identified graphically through box plots and were only removed when confirmed through Grubbs's tests (OriginPro). For comparisons of means between three or more independent groups, Tukey test ANOVA for multiple comparison was performed. For correlation analysis, Spearman's rank correlation was performed. Pearson's correlation coefficient was calculated for measuring the association between variables of interest based on the method of covariance which also gives information about the magnitude of the association, or correlation, as well as the direction of the relationship. p-values<0.05 for correlations or means' comparisons were considered significant. Categorical variables are reported as counts and percentages while continuous variables are expressed as mean ± standard deviation. Differences between percentages were assessed by Pearson's $\chi^2$ tests or Fisher's exact tests when the number of observations per group was less than 5. The $\chi^2$ tests provided results that tested the hypothesis that the mortality and a given variable (e.g., sex or comorbidity) are independent. When p is less than the significant level of 0.05, there is significant evidence of association between mortality and the variable. Log-rank Kaplan–Meier survival analyses were carried out to estimate the probability of survival of COVID-19 patients in relation to cutoff thresholds arbitrarily stratified at mean values of various parameters. The log-rank test for trends reports a $\chi^2$-value and computes a p value testing the null hypothesis that there is no linear trend between column order and median survival.

## Acknowledgements

The present work was funded by the Association of Friends of the National Cancer Institute and the Children's Cancer Hospital Foundation.

## Additional information

### Funding

| Funder | Grant reference number | Author |
| --- | --- | --- |
| The Association of Friends of the National Cancer Institute | COVID-SA | Sameh Saad Ali |
| The Children's Cancer Hospital Egypt | SA-Start up | Sameh Saad Ali |

| Funder | Grant reference number | Author |
|--------|------------------------|--------|

The funders had no role in study design, data collection and interpretation, or the decision to submit the work for publication.

## Author contributions

Mohamed A Badawy, Basma A Yasseen, Aya A Elkhodiry, Azza G Kamel, Data curation, Formal analysis, Visualization, Writing – review and editing; Riem M El-Messiery, Conceptualization, Data curation, Project administration, Supervision, Writing – review and editing; Engy A Abdel-Rahman, Conceptualization, Data curation, Formal analysis, Methodology, Supervision, Writing – review and editing; Hajar El-sayed, Data curation; Asmaa M Shedra, Rehab Hamdy, Data curation, Writing – review and editing; Mona Zidan, Diaa Al-Raawi, Data curation, Formal analysis, Writing – review and editing; Mahmoud Hammad, Mohamed El Ansary, Ahmed Al-Halfawy, Ashraf Hatem, Resources, Writing – review and editing; Nahla Elsharkawy, Methodology, Resources, Writing – review and editing; Alaa Elhadad, Conceptualization, Writing – review and editing; Sherif Abouelnaga, Funding acquisition, Project administration, Resources, Writing – review and editing; Laura L Dugan, Conceptualization, Validation, Writing – review and editing; Sameh Saad Ali, Conceptualization, Data curation, Formal analysis, Funding acquisition, Investigation, Methodology, Project administration, Resources, Supervision, Visualization, Writing - original draft

## Author ORCIDs

Mohamed A Badawy http://orcid.org/0000-0003-1691-0167
Aya A Elkhodiry http://orcid.org/0000-0001-5684-0242
Mahmoud Hammad http://orcid.org/0000-0003-1677-0360
Sameh Saad Ali http://orcid.org/0000-0002-0339-6106

## Ethics

Human subjects: Written informed consents were obtained from participants in accordance with the principles of the Declaration of Helsinki. For COVID-19 and control blood/plasma collection, Children's Cancer Hospital's Institutional Review Board (IRB) has evaluated the study design and protocol, IRB number 31-2020 issued on July 6, 2020.

## Decision letter and Author response

Decision letter https://doi.org/10.7554/eLife.69417.sa1
Author response https://doi.org/10.7554/eLife.69417.sa2

# Additional files

## Supplementary files

- Supplementary file 1. Calculated biophysical EPR spectral parameters for all groups.
- Transparent reporting form

## Data availability

All data generated or analysed during this study are included in the manuscript and supporting files. Raw data collected and used to produce all figures and tables are available on Dyrad (https://doi.org/10.5061/dryad.cnp5hqc4q).

The following dataset was generated:

| Author(s) | Year | Dataset title | Dataset URL | Database and Identifier |
|-----------|------|---------------|-------------|-------------------------|
| Ali SS | 2021 | Data from: Biophysical Data Pertaining to COVID-19 caused human serum albumin damage | https://doi.org/10.5061/dryad.cnp5hqc4q | Dryad Digital Repository, 10.5061/dryad.cnp5hqc4q |

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
