## [Editor Report]

This submission is novel since it provides information on the structure changes of albumin in COVID-19.

---

## [Decision Letter]

**Decision letter after peer review:**

Thank you for submitting your article "Neutrophil-mediated Oxidative Stress and Albumin Structural Damage Predict COVID-19-associated Mortality" for consideration by *eLife*. Your article has been reviewed by 3 peer reviewers, including Evangelos J Giamarellos-Bourboulis as the Reviewing Editor and Reviewer #3, and the evaluation has been overseen by Y M Dennis Lo as the Senior Editor.

Essential revisions:

• The authors state that HSA is modified, and this may influence the normal functioning of the protein. Can the authors provide evidence that such modifications will result in ablation of normal HSA functions using orthogonal in vitro experiments?

• Although neutrophils can produce hydrogen peroxide via the NADPH oxidase-superoxide dismutase axis, mitochondria are even more ubiquitous and can produce hydrogen peroxide in higher amounts. Therefore, the role of mitochondria in adding to hydrogen peroxide pool in plasma should be discussed.

• Reference for lysis of blood to isolate neutrophils for further use in DCF based ROS detection using FACs is missing. FACS lyse and erythrolyse solutions used for RBC can modify neutrophil membranes and enhance the DCF signal. The authors expose neutrophils to the lysis buffer for 15 minutes and this is enough to activate the neutrophils. This may have introduced false positives. Also, if this was a lyse-fix solution a similar issue of activation and a false positive signal may be encountered.

• Has cellular viability been assessed for the neutrophils after the lysis? This information should be provided.

• Also, in Figure 3D, the nuclei of the control images have their characteristic polymorphonuclear appearance and appear as round. This indicates that the neutrophils may have been lysed and may be due to the harsh RBC lysis based neutrophil isolation method.

• ROS species produced by neutrophils is comprised of hydrogen peroxide and the superoxide free radical. Can the authors discuss as to why only peroxide levels were chosen?

• How were the neutrophils isolated and how many cells seeded used for the fluorescence imaging? There is not enough information if the described structural changes are found in other infectious processes or if they are unique to COVID-19.

• The authors need to elaborate more on the importance of the findings regarding pathogenesis. They cannot analyze for survival with only 25 patients.

• How was the study powered for?

• Do albumin alteration associate with other immune changes of the patients?

• Why are not other markers of oxidative damage measured like malondialdehyde and total antioxidant?

• The authors exaggerate by stating in the text as significant differences that are quoted as marginally different in the Figures.

• In figure 2 the authors mention a population of neutrophils as 'NETosed'. Can the authors elaborate as to what this means? If they mean NET formation then they should mention what parameters were considered to define NET formation or exclude nomenclature from the figure as it is confusing.

• In Figure 3D depicting DCF in neutrophils, the Hoechst 332 staining in the merged control image looks extremely faint compared to the Sev-R and Sev-D merged images. Higher quality images should be provided for the panel.

• Could the authors validate the efficacy of the combined S/W/H_2_O_2_ versus other established markers of mortality in COVID-19 such as D-dimer, Ferritin etc.

• Figure-2 A FACS plots should be labelled within the figure for the ease of reader.

• DAPI mentioned in Figure 3 is equated to Hoechst 332 in the methods section. This should be rectified.

---

## [Author Response]

Essential revisions:• The authors state that HSA is modified, and this may influence the normal functioning of the protein. Can the authors provide evidence that such modifications will result in ablation of normal HSA functions using orthogonal in vitro experiments?

We agree that this is indeed an important point and thus performed new set of experiments to explore the impact of COVID-19 on HSA transport function as reflected in rates of fatty acid uptake and release. Before describing these new results we would like to mention that the restricted water/ascorbate accessibility in severe patients relative to controls (Figure 4I and J) along with the observed significant changes in the microenvironments surrounding SLFAs are generally viewed to implicate functional damage [PMIDs: 213518,2155659,19540798,25152968,20886483]. In this paradigm changes in the SLFA’s lineshape are taken to indirectly display functional changes in the solution shape of albumin due to spatial rearrangements of paramagnetic centers in EPR-active, albumin-bound fatty acids.

However, to confirm that the observed protein damages are indeed functional determinants for HSA we performed additional experiments exploring albumin transport functions through loading/unloading of SLFA for representative sets of samples (n=3-5) from all of the studied groups. We argued that if the observed HSA structural damages are functionally relevant they would affect the ability of HSA to load and dislodge fatty acids into and from their binding sites [PMID: 12350048]. To follow the fatty acid loading efficacy, we followed the apparent kinetics of 16-DSA’s S (New Figure 5A; strongly bound) and W (New Figure 5B; weakly bound) peaks over 7 minutes immediately after mixing the spin label with identical plasma volumes from all groups. These results (N=3) demonstrate the hindered fatty acid uptake by HSA of critically-ill patients relative to controls.

To investigate the dislodging function of HSA we monitored the effect of exogenously added ethanol on S and W peaks’ proportions in all groups. Namely, we compared the amounts of exogenous ethanol needed to withdraw the weakly and strongly bound components of the HSA-bound SLFAs in whole blood (New Figure 5D and E). Under these conditions, the W peak envelops contributions from both weakly bound and the released SLFA. Redistributions of the fatty acid populations are noticeable through decreased S and increased W (includes signals of free fatty acid) peaks. This redistribution is remarkably pronounced in critically ill patients reflecting weaker association with, and easier release of fatty acids from HSA in those patients. These findings are taken to indicate impaired transport function of HSA in critically ill patients. The new data are added as an additional figure (Figure 5) in the resubmitted manuscript.

• Although neutrophils can produce hydrogen peroxide via the NADPH oxidase-superoxide dismutase axis, mitochondria are even more ubiquitous and can produce hydrogen peroxide in higher amounts. Therefore, the role of mitochondria in adding to hydrogen peroxide pool in plasma should be discussed.

Changes inflected by COVID-19 pathology on mitochondrial function including oxidative phosphorylation, calcium handling, and ROS production in platelets and neutrophils are described in a manuscript to be submitted soon. Results obtained through multiple experimental approaches including flowcytometry, fluorescence imaging, and fluorescence microscopy suggest that mitochondrial ROS are independent of severity and mortality outcomes. However, we added a few sentences to discuss the role of mitochondria as suggested (page 14; Lines: 309-312).

• Reference for lysis of blood to isolate neutrophils for further use in DCF based ROS detection using FACs is missing. FACS lyse and erythrolyse solutions used for RBC can modify neutrophil membranes and enhance the DCF signal. The authors expose neutrophils to the lysis buffer for 15 minutes and this is enough to activate the neutrophils. This may have introduced false positives. Also, if this was a lyse-fix solution a similar issue of activation and a false positive signal may be encountered.

We have mentioned in the method’s section that “Citrate-anticoagulated whole peripheral blood was incubated with RBCs lysis buffer for 15 min. Lysed blood was then centrifuged at 500 X-g for 5 min. Cells were washed twice with phosphate buffered saline (PBS) and then resuspended in PBS.”

We didn’t use FACS lyse and erythrolyse solutions in our experiments, for lysis of blood we used RBS lysis buffer that consists of: NH_4_Cl (ammonium chloride), NaHCO_3_ (sodium bicarbonate) and EDTA (disodium). This lysis buffer is known to preserve viability of cells and is the buffer of choice for assessment of neutrophil respiratory burst using DCF (PMID:11260596). Moreover, we have suspended the obtained cells after lysis in PBS and we avoided the use of FACS as it contains FBS or BSA that may cause activation of neutrophils. Methods’ Section was modified to describe this {page 20, Lines 411-420}.

• Has cellular viability been assessed for the neutrophils after the lysis? This information should be provided.

Cellular viability has always been checked for each sample by trypan blue and cells were not analysed unless it was more than 90% viable. Added in the resubmitted manuscript {page 22, lines: 460-461}.

• Also, in Figure 3D, the nuclei of the control images have their characteristic polymorphonuclear appearance and appear as round. This indicates that the neutrophils may have been lysed and may be due to the harsh RBC lysis based neutrophil isolation method.

We have now included a better representative control images in Figure 3D. As mentioned above, the utilized lysis buffer and conditions are known to preserve viability of neutrophil (PMID:11260596). Moreover, we have suspended the obtained cells after lysis in PBS and we avoided the use of FACS as it contains FBS or BSA that may cause activation of neutrophils. Cellular viability has always been checked for each sample by trypan blue and it was more than 90%.

• ROS species produced by neutrophils is comprised of hydrogen peroxide and the superoxide free radical. Can the authors discuss as to why only peroxide levels were chosen?

We agree that superoxide is as important as hydrogen peroxide in this context. However, we focused on hydrogen peroxide because:

1. Hydrogen peroxide is a direct product of superoxide dismutation (auto and enzymatic).

2. The superoxide radical anion is very short-lived and much harder to detect than hydrogen peroxide.

3. Hydrogen peroxide is the most stable ROS and is barely affected under prolonged storage at low temperatures.

4. The activities of all enzymatic sources of superoxide (e.g. NOXs) are time sensitive thus requiring to handle samples immediately, which was not always possible. In our accumulated experience, discrepancies in sample handling times are remarkable sources of experimental errors in detecting superoxide radicals.

5. Methods for the determination of superoxide radical are not quantitative while the described catalase-based assay determines hydrogen peroxide quantitatively.

6. Except for EPR spectroscopy, superoxide detection methods are non-specific and suffer from interferences especially in clinical samples.

• How were the neutrophils isolated and how many cells seeded used for the fluorescence imaging?

Here is the method followed to isolate neutrophils and the employed seeding density:

Whole blood samples (10ml) were collected on ACD tubes and transferred to 15 mL polypropylene conical tubes then centrifuged (300xg) for 15 minutes with reduced acceleration and no brakes. The upper layer (PRP) was transferred to a separate tube. The lower layer was layered over equal volume of Histopaque-1077. Then the tubes were centrifuged at 500xg for 35 minutes with reduced acceleration and no brakes. After centrifugation, the first three layers were discarded and the layer containing the neutrophils was collected and diluted to 10 mL with HBSS, centrifuged again at 350xg for 10 minutes. Supernatant was removed and the pellet was suspended in RBCs lysis buffer and incubated for 15 min at RT then centrifuged at 350xg for 10 minutes (Abdel-Rahman et al., 2021; PMID: 33402007). The supernatant was removed and the pellet was suspended in 100 µL of PBS. The count was obtained manually using a hemocytometer. For the fluorescence imaging using cytation 5, neutrophils were seeded in a flat clear bottom 96 well plate in a density of 100,000 neutrophils/ well. Added to the Methods’ Section {page 21, lines 449-460}.

There is not enough information if the described structural changes are found in other infectious processes or if they are unique to COVID-19.

Although hypoalbuminemia is suggested as a pivotal marker for the pathology of many infections [reviewed very recently in 33925831] including in COVID-19 [ref 33], albumin damage has not been directly implicated in infectious disease to the best of our knowledge. Added in the Discussion section {page 14, lines 323-325}.

• The authors need to elaborate more on the importance of the findings regarding pathogenesis.

We added the following paragraph to the resubmitted manuscript {last paragraph on page 14}:

“Oxidative stress is commonly implicated in a plethora of diseases and was indeed suggested repeatedly; albeit with little experimental support, as the spearhead of inflammation leading to severe symptoms and sepsis-like fatal responses in critically ill COVID-19 patients (15). […] Collectively, these factors constitutes major players in the pathology triggered by SARS-CoV-2 viral infection.”

They cannot analyze for survival with only 25 patients.

We have now increased the number of subjects to be 39 instead of 25. The current survival analysis reports event numbers in the range from 15 to 21 for various predictor parameter (see new Figure 6G-J). This exceeds current consensus that 10 or more events per predictor are sufficient to provide firm grounds for generalization of findings [PMID: 21411281, 24115780, 26964707]. In fact, when we increased the number of subjects from 25 to 39 mean values of the studied parameters were marginally affected while maintaining their impact on survival outcomes. Added in the Discussion section {page 14, lines 306-309}.

• How was the study powered for?

As stated in the manuscript {pages 17, line 353-355}, no power analysis was done due to lack of reported effect size under matching or similar conditions. Sample size was based on sample availability and collection continued until a total of N=40 has been reached for COVID-19 patients diagnosed as severely infected with SARS-CoV-2 and admitted to the ICU during the period from 13 October 2020 to 21 February 2021. Nevertheless, and as explained in the previous point, our expanded survival analysis contains sufficiently powered sample size.

• Do albumin alteration associate with other immune changes of the patients?

As indicated in Figure 2B, neutrophil and ROS-positive neutrophil counts increased remarkably in association with severity and mortality. This increase was strongly correlated with parameters reflecting albumin damage (Figure 2C and 6A-F). Lymphocytes and platelets however showed the opposite trend thus potentiating the centrality of neutrophils in the COVID-19 triggered inflammation and oxidative stress. Detailed immune profiling of the studied patients was beyond the current study goals. However, we believe that this is an interesting suggestion that deserves more exploration provided that sample availability returns.

The current manuscript was more focused on the association of albumin damage with inflammatory markers. We believe that the current data suggest that albumin-related changes are more robust predictors of mortality and severity in the studied cohort. That is, all immune-related parameters such as cytokine levels, CRP, D-dimer, etc. were not statistically different when comparing survivors with non-survivors (Table 2). Nevertheless, obtained biophysical and biochemical parameters correlated reasonably well with severity and mortality. Added in the Discussion section {page 15, lines 326-329}.

• Why are not other markers of oxidative damage measured like malondialdehyde and total antioxidant?

Because we reasoned that highly uncontrolled intake of antioxidants such as ascorbic acid and zinc by COVID-19 patients may lead to huge experimental variability and clinically irrelevant conclusions in case of the total antioxidant assessments. For other markers of oxidative stress such as malondialdehyde our plans were hampered by the lockdown and inability to import reagents in a timely manner.

• The authors exaggerate by stating in the text as significant differences that are quoted as marginally different in the Figures.

We understand this comment and have revised our manuscript to maintain a conservative descriptions of our data.

• In figure 2 the authors mention a population of neutrophils as 'NETosed'. Can the authors elaborate as to what this means? If they mean NET formation then they should mention what parameters were considered to define NET formation or exclude nomenclature from the figure as it is confusing.

We apologize for this mistake. We masked the label in the resubmitted Figure 2 as it is irrelevant to the current manuscript.

• In Figure 3D depicting DCF in neutrophils, the Hoechst 332 staining in the merged control image looks extremely faint compared to the Sev-R and Sev-D merged images. Higher quality images should be provided for the panel.

We apologize for the bad quality of the formerly provided images. We have now included images with higher quality and clearer nuclear staining.

• Could the authors validate the efficacy of the combined S/W/H_2_O_2_ versus other established markers of mortality in COVID-19 such as D-dimer, Ferritin etc.

We couldn’t find statistically significant correlations with any of these parameters. It is worth mentioning that analyses of these markers were carried out in the ICU at times not necessarily matching the times when we collected samples for our analysis. We speculate that the time variability of these parameters renders such analysis inconclusive.

• Figure-2 A FACS plots should be labelled within the figure for the ease of reader.

To avoid readers’ confusion and over-labeling of the figure we color-framed individual rows to distinguish Control, Sev-R, or Sev-D panels.

• DAPI mentioned in Figure 3 is equated to Hoechst 332 in the methods section. This should be rectified.

We apologize for this mistake, which was corrected in the resubmitted Figure 3.